# Prognostic Value of Dynamic Changes in Immune-Inflammatory and Tumor Biomarkers Following Chemoradiotherapy in Locally Advanced Rectal Cancer

**DOI:** 10.3390/cancers17203383

**Published:** 2025-10-20

**Authors:** Mahmoud Al-Masri, Yasmin Safi, Mohammad Almasri, Ramiz Kardan, Daliana Mustafa, Osama Alayyan, Bilal Kahalah, Rama AlMasri

**Affiliations:** 1Department of Surgery, King Hussein Cancer Center, Amman 11941, Jordan; ys.11060@khcc.jo (Y.S.); mohammadalmasri1998@gmail.com (M.A.); ramiz.kardan@gmail.com (R.K.); oa.16891@khcc.jo (O.A.); bilal.kahhaleh@gmail.com (B.K.); rmalmasri97@gmail.com (R.A.); 2Faculty of Medicine, University of Jordan, Amman 11942, Jordan; 3Office of Research and Affairs, King Hussein Cancer Center, Amman 11941, Jordan; dalianamustafa@gmail.com

**Keywords:** rectal cancer, neutrophil-to-lymphocyte ratio, platelet-to-lymphocyte ratio, CEA, immunoinflammatory biomarkers, chemoradiotherapy, surgery, survival analysis, predictive markers

## Abstract

Patients with locally advanced rectal cancer (LARC) who receive neoadjuvant chemoradiotherapy followed by surgery have variable outcomes, and identifying those at higher risk of recurrence remains a clinical challenge. Current tools for assessing prognosis rely largely on static clinical or pathological features, which may not fully capture individual risk. This study investigates dynamic changes in blood-based markers, specifically neutrophil-to-lymphocyte ratio (NLR) and carcinoembryonic antigen (CEA), to evaluate their ability to predict overall and disease-free survival in LARC patients. We found that patients with reductions in both markers after treatment had better survival, while increases were associated with worse outcomes, and combining the markers improved prognostic accuracy. These findings suggest that monitoring changes in NLR and CEA could help guide personalized treatment and follow-up strategies for rectal cancer patients.

## 1. Introduction

Colorectal cancer represents a significant clinical challenge and remains the second leading cause of cancer-related mortality and the third most common cancer worldwide [1]. In 2020, GLOBOCAN estimated over 1.9 million new cases and more than 930,000 deaths globally. By 2040, the burden is projected to increase to 3.2 million new cases and 1.6 million deaths [2]. In Jordan, it ranks as the second most prevalent cancer among both genders, highlighting its public health importance both locally and globally [3].

Rectal cancer constitutes approximately one-third of all colorectal cancers [4]; of these, up to 35% presented with locally advanced rectal cancer (LARC) at the time of diagnosis [5]. The current treatment approach of locally advanced rectal cancer (LARC) is multidisciplinary, including neoadjuvant radiotherapy or chemoradiotherapy before resection in the total mesorectal excision (TME) with or without postoperative adjuvant chemotherapy [6]. This multimodal approach improves loco regional control of LARC, although overall survival (OS) remains unchanged. This could be attributed to low tolerance or compliance of most LARC patients to complete the started adjuvant chemotherapy [7]. Recent research has focused on the potential therapeutic benefits of total neoadjuvant therapy (TNT), which has shown improvement in short and long-term oncological outcomes, including distant metastases-free survival and overall survival, and patient treatment compliance [7,8]. This has led to international guidelines including TNT as the standard of care for LARC meeting specific criteria [9]. However, survival outcomes remain heterogeneous, underscoring the need for reliable prognostic markers to guide individualized treatment and follow-up strategies.

Solid tumor is characterized by heterogeneity at molecular levels [10]. Such heterogeneity mandates the development of risk prediction tools to help physicians in risk stratification, treatment planning, follow-up scheduling and patients’ counselling. The current prediction tools are based on clinicopathological features alone with limited prediction accuracy [11]. Therefore, integration of immunoinflammatory biomarkers appears to improve the prediction accuracy of these models, and this has been demonstrated in several malignancies such as breast, prostate and bladder cancers [12].

Systemic inflammatory response markers such as the neutrophil-to-lymphocyte ratio (NLR) and platelet-to-lymphocyte ratio (PLR) have emerged as promising prognostic indicators in various solid tumors, including colorectal cancer [13,14,15]. These biomarkers are readily available, cost-effective, and reflect the interplay between tumor biology and host immune response [16]. Despite their individual utility, limited studies have evaluated the combined and dynamic changes in these markers—particularly pre- and post-CRT—in predicting long-term outcomes in rectal cancer patients [17].

Moreover, carcinoembryonic antigen (CEA) is one of the most widely used tumor markers in colorectal cancer and plays a key role in both diagnosis and disease monitoring [18,19]. CEA is a glycoprotein involved in cell adhesion and is normally present at low levels in adult serum, but its expression is elevated in many malignancies, particularly adenocarcinomas of the gastrointestinal tract [18]. In rectal cancer, pre-treatment CEA levels have been consistently associated with tumor burden, lymph node involvement, and advanced stage. Elevated baseline CEA has been shown to correlate with poorer overall survival (OS) and disease-free survival (DFS), making it a valuable prognostic biomarker [20]. Moreover, changes in CEA levels after neoadjuvant chemoradiotherapy (CRT) may reflect treatment response and residual tumor activity [21]. Studies have demonstrated that patients with persistently high or rising CEA levels post-CRT are at increased risk of recurrence and mortality, supporting its role in postoperative risk stratification and surveillance [22]. Due to its accessibility, reproducibility, and clinical utility, CEA remains an essential component of prognostic assessment in rectal cancer management [23,24].

Recently, growing interest has emerged in combining CEA with systemic inflammatory markers such as the neutrophil-to-lymphocyte ratio (NLR) to improve prognostic accuracy. While each marker individually provides moderate predictive value, their combination reflects both tumor biology and the host immune response [25,26]. This study aims to explore the prognostic value of pre- and post-treatment CEA, NLR and PLR in patients with LARC undergoing neoadjuvant therapy followed by surgery. Specifically, we assess their association with overall survival (OS) and disease-free survival (DFS), examine cut-off values derived from ROC curve analysis, evaluate the impact of biomarker changes (delta values), and explore their combined predictive power.

## 2. Methods

### 2.1. Study Design and Population

This retrospective cohort study included patients diagnosed with locally advanced rectal cancer (LARC) who received neoadjuvant CRT as CCRT or TNT, followed by curative-intent surgery at King Hussein Cancer Center (KHCC) between 2014 and 2024. During this study period, two treatment protocols were implemented. From 2014 until the end of 2020, patients were treated with standard long-course concurrent chemoradiotherapy. Starting in 2021, institutional guidelines were updated to adopt total neoadjuvant therapy (TNT).

Eligible patients were ≥18 years old with histologically confirmed rectal adenocarcinoma, clinical stage II or III disease, underwent staging with MRI prior to CRT and surgery, and available laboratory data on neutrophil-to-lymphocyte ratio (NLR), and platelet-to-lymphocyte ratio (PLR), both before and after CRT. For patients treated with conventional concurrent chemoradiotherapy (CCRT), completion of the planned postoperative adjuvant chemotherapy regimen was required for inclusion. Patients with early-stage (T1/T2, node-negative), metastatic disease, synchronous/previous malignancies or prior pelvic radiotherapy, inflammatory conditions, clinical evidence of sepsis, tumor perforation, or incomplete/palliative surgery or incomplete clinical data were excluded.

In addition, demographic characteristics and comorbidities were reviewed to ensure representativeness of the study cohort and to capture potential confounders in survival analyses. Imaging reports and pathology reports were reviewed to confirm staging, tumor response, and surgical margins. All patients were managed according to standardized institutional protocols to minimize variability in treatment and follow-up.

### 2.2. Treatment Protocol

All patients received neoadjuvant therapy followed by total mesorectal excision (TME) surgery, in accordance with institutional protocols. Between 2014 and 2020, patients were treated with standard long-course concurrent chemoradiotherapy (CCRT), consisting of pelvic radiotherapy at a total dose of 45–50.4 Gy delivered in 25–28 fractions, administered concurrently with fluoropyrimidine-based chemotherapy (5-fluorouracil). Surgery was performed 4–6 weeks after completion of CRT, followed by adjuvant chemotherapy using either FOLFOX for at least 8 cycles or XELOX from 6 cycles regimens.

Starting in 2021, total neoadjuvant therapy (TNT) became the standard protocol. Patients received short-course radiotherapy (25 Gy in 5 fractions), followed by systemic chemotherapy consisting of either six cycles of XELOX or nine cycles of FOLFOX. TME surgery was performed 4–6 weeks after the completion of neoadjuvant chemotherapy. Surgical procedures included low anterior resection (LAR), abdominoperineal resection (APR), or pelvic exenteration, depending on tumor location, extent of the disease, response to treatment, and resectability.

### 2.3. Data Collection

Clinical and pathological data were retrieved from electronic medical records and institutional cancer registries. Baseline variables included age, gender, BMI, smoking status, comorbidities, clinical stage, surgical type, and pathological American Joint Committee on Cancer (AJCC) stage. Laboratory data included pre- and post-CRT treatment values for complete blood count (CBC) and carcinoembryonic antigen (CEA). CBCs were used to calculate inflammatory biomarkers, including the neutrophil-to-lymphocyte ratio (NLR) and platelet-to-lymphocyte ratio (PLR). NLR was calculated as the ratio of the absolute neutrophil count to the absolute lymphocyte count, and PLR was calculated as the ratio of the platelet count to the absolute lymphocyte count. These values were recorded at two standardized time points: (1) prior to the initiation of chemoradiotherapy and (2) within 2–4 weeks after completion of neoadjuvant therapy but before surgery (See Figure A1).

Serum CEA levels were also measured at these two time points. According to institutional guidelines, a CEA level of 5 ng/mL was used as the threshold to define elevated versus normal values. These biomarker measurements were used for both static analysis and to assess dynamic changes (delta values) between pre- and post-treatment, allowing for their association with survival outcomes to be evaluated. All collected clinical, pathological, and laboratory data were cross-verified by two independent researchers to ensure accuracy and completeness.

### 2.4. Outcomes and Follow-Up

The primary outcomes were overall survival (OS) and disease-free survival (DFS). OS was defined as the time from the date of diagnosis to death, and DFS as the time from diagnosis to disease recurrence or death, whichever occurred first. Patients were followed through routine clinic visits and imaging, and survival data were censored at the last known follow-up.

### 2.5. Statistical Analysis

All laboratory values for NLR, PLR, and CEA were inspected for extreme outliers prior to analysis using the interquartile range (IQR) method applied to log-transformed values. No values were excluded, as all measurements were within physiologically plausible ranges and reflected true biological variability among patients. To ensure robustness, sensitivity analyses were additionally performed after excluding extreme outliers, with results compared to the full dataset.

Continuous variables were summarized using mean and standard deviation (SD) or median and interquartile range (IQR) as appropriate. Categorical variables were reported as frequencies and percentages. For paired comparisons of biomarker levels (NLR, PLR, and CEA) before and after chemoradiotherapy (CRT), we first assessed the distribution of paired differences. When the assumption of normality was met, paired t-tests were applied; otherwise, the non-parametric Wilcoxon signed-rank test was used. Results from both tests are reported.

For survival analyses, biomarkers were evaluated both as continuous variables and as categorical groups. Kaplan–Meier survival analysis was performed to evaluate OS and DFS according to categorical biomarker groups, with log-rank tests used for comparisons. Optimal cut-off values for post-CRT NLR were determined using receiver operating characteristic (ROC) curve analysis, with the Youden index used to identify the threshold that maximized sensitivity and specificity. For CEA, a cut-off value of 5 ng/mL was used based on established literature and internal clinical guidelines.

Patients were stratified into delta groups based on the direction of change in biomarker levels pre- and post-CRT (e.g., increased vs. decreased NLR and PLR, or combinations thereof). Survival differences across these delta groups were assessed using Kaplan–Meier curves.

Cox proportional hazards models were used to perform univariable and multivariable survival analyses. Variables with a *p*-value < 0.05 in univariable analysis were included in the multivariable models. Differences in treatment regimens, including the transition from long-course concurrent chemoradiotherapy with adjuvant chemotherapy (2014–2020) to total neoadjuvant therapy (2021 onward), were evaluated in univariable analysis. Variables showing a significant association with survival in univariable models were included as covariates in multivariable Cox proportional hazards models to adjust for potential confounding effects. Hazard ratios (HRs) with 95% confidence intervals (CIs) and *p*-values were reported. Statistical significance was set at *p* < 0.05. All analyses were performed using R statistical software version 4.2.1 (R Foundation for Statistical Computing, Vienna, Austria).

### 2.6. Ethical Considerations

The study was approved by the Institutional Review Board (IRB) of KHCC (IRB number: 21 KHCC 07), and all procedures adhered to the ethical standards of the Declaration of Helsinki. Informed consent was waived due to the retrospective nature of the study.

## 3. Results

### 3.1. Patient Demographics and Clinical Characteristics

A total of 591 patients with histologically confirmed rectal cancer were initially identified. Of these, 312 patients were classified as having clinical stage II or III disease. After applying exclusion criteria, 43 patients were excluded due to missing data, and 8 patients did not undergo surgery. The final study cohort included 261 patients who received neoadjuvant chemoradiotherapy followed by curative-intent surgery. One hundred and sixty-nine patients (64.8%) received concurrent chemoradiotherapy (CCRT), subsequently underwent surgery, and completed the full indicated adjuvant chemotherapy, where 92 patients (35.2%) received TNT then underwent surgery (Figure A1).

The mean age was 55.5 years (SD = 11.4), ranging from 25 to 80 years. In addition, 55.9% of the patients were male, and the average body mass index (BMI) was 29.5 (SD = 5.9). Comorbidities were present in 62.5% of patients, and 37.5% were current or former smokers. Clinical staging revealed that 93.5% had stage III disease. Regarding surgical procedures, 71.6% underwent low anterior resection (LAR), 27.2% abdominoperineal resection (APR), and 1.1% pelvic exenteration. Overall, 61.7% of surgeries were performed via an open approach, while 38.3% were minimally invasive procedures. Postoperative AJCC pathologic staging showed 36.4% of patients were stage II and 28.7% stage III. Postoperative complications were reported in 39.1% of patients (Table 1).

### 3.2. Changes in Inflammatory and Tumor Markers Pre- and Post-CRT

The distributions of NLR, PLR, and CEA were non-normal, as assessed by the Shapiro–Wilk test. There were significant increases in inflammatory and tumor markers following CRT (Table 2 and Figure 1). The mean NLR increased from 3.2 (SD = 2.1) pre-CRT to 4.9 (SD = 4.3) post-CRT (*p* < 0.0001). Similarly, PLR rose from 185.4 (SD = 110.2) to 293.5 (SD = 217.6) post-CRT (*p* < 0.0001). While mean CEA levels appeared to increase from 11.4 ng/mL (SD = 20.2) pre-CRT to 20.6 ng/mL (SD = 95.8) post-CRT, this change was not statistically significant (*p* = 0.126). However, the Wilcoxon signed-rank test, which accounts for the non-normal distribution, indicated a significant increase in CEA levels post-CRT (*p* < 0.01) (Table 2).

### 3.3. Univariable and Multivariable Survival Analysis of Clinical and Biomarker Variables

#### 3.3.1. Overall Survival (OS)

In univariable analysis, comorbidity (HR = 3.49, *p* = 0.004), AJCC pathologic stage III (HR = 3.67, *p* = 0.036), age (HR = 1.03, *p* = 0.011), post-CRT CEA (HR = 1.00, *p* < 0.001), and post-CRT NLR (HR = 1.06, *p* = 0.007) were associated with OS. In the multivariable model, both post-CRT CEA (*p* < 0.001) and post-CRT NLR (*p* = 0.007) remained statistically significant, while comorbidity and AJCC stage III showed a trend toward significance (Table 3).

#### 3.3.2. Disease-Free Survival (DFS)

Univariable predictors of DFS included AJCC pathologic stage III (HR = 5.10, *p* = 0.002) and post-CRT CEA (HR = 1.00, *p* < 0.001). Both variables remained significant in the multivariable analysis, confirming their independent prognostic value for recurrence (Table 3).

### 3.4. ROC Analysis and Biomarker Cut-Offs

Receiver operating characteristic (ROC) analysis identified an optimal cut-off point of 8.807 for post-CRT NLR in predicting overall survival (Figure 2). This threshold was subsequently used to classify patients into high and low NLR groups for survival analysis (Figure 2).

### 3.5. Kaplan–Meier Survival Stratified by NLR

Kaplan–Meier curves demonstrated significantly worse overall and disease-free survival for patients with high post-CRT NLR (>8.807) compared to those with lower values, reinforcing the prognostic role of post-treatment inflammation (Figure 3).

### 3.6. Prognostic Role of Delta NLR and PLR Combinations

Patients were classified into four delta groups based on changes in NLR and PLR. Kaplan–Meier analysis showed that Group upNLR and downPLR had the worst OS and DFS, while Group downNLR and downPLR had the most favorable outcomes. Groups with discordant changes had intermediate, comparable survival, suggesting that changes in NLR exert greater prognostic influence than PLR (Figure 4).

### 3.7. Combined Predictive Performance of Post-CRT CEA and NLR

To assess the combined predictive performance of post-CRT CEA and NLR, ROC analysis was conducted. CEA alone yielded an AUC of 0.70, and NLR alone an AUC of 0.66. When combined, the predictive accuracy improved markedly, with an AUC of 0.84, underscoring their complementary value in stratifying mortality risk (Figure 5).

### 3.8. Survival Impact of NLR and CEA Groups

Using cut-offs of >8.807 for NLR and >5 ng/mL for CEA (CEA cut-off was defined as 5 ng/mL based on existing literature and institutional guidelines [27]), patients were grouped accordingly. Multivariable Cox analysis revealed that high post-CRT NLR was associated with significantly worse OS (HR = 3.62; 95% CI: 1.82–7.20; *p* < 0.001), as was high post-CRT CEA (HR = 15.11; 95% CI: 7.98–28.61; *p* < 0.001). These results confirm the independent prognostic value of both markers (Figure 6).

### 3.9. Prognostic Value of the Dynamic Changes in CEA and NLR

Further Kaplan–Meier analysis assessed dynamic changes in NLR and CEA pre- vs. post-CRT. Patients with reductions in both markers had the best survival outcomes, while those with elevations in both had significantly poorer OS and DFS (*p* < 0.0001 and *p* = 0.00019, respectively). These findings reinforce the prognostic relevance of biomarker dynamics over static measurements (Figure 7).

## 4. Discussion

This study demonstrated that dynamic changes in inflammatory and tumor markers, specifically neutrophil-to-lymphocyte ratio (NLR) and carcinoembryonic antigen (CEA), have significant prognostic value in patients with locally advanced rectal cancer (LARC) undergoing neoadjuvant chemoradiotherapy (CRT) followed by surgery. In our study, we showed that post-CRT NLR was significantly associated with OS (HR = 1.06 (95% CI: 1.02–1.10, *p* = 0.003)). Specifically, an NLR cut-off of 8.807 was identified using ROC analysis, above which patients’ OS was significantly poorer, and this was also confirmed using Kaplan–Meier analysis outcomes (HR = 1.06; 95% CI: 1.02–1.10; *p* = 0.003). Similarly, post-CRT CEA emerged as a strong independent predictor for both OS and DFS (*p* < 0.001), with ROC analysis showing an AUC of 0.70 for CEA alone. Notably, when combined with NLR, the predictive accuracy improved markedly (AUC = 0.84), underscoring their complementary prognostic value. Kaplan–Meier analysis further revealed that patients with reductions in both CEA and NLR had the most favorable survival, while those with elevations in both demonstrated the poorest OS and DFS (*p* < 0.0001 and *p* = 0.00019, respectively). However, combined delta group analysis of NLR and PLR revealed unique survival outcomes. Patients characterized by an increase in NLR and a decrease in PLR and increase in both NLR and PLR following chemoradiotherapy, respectively, demonstrated the worst overall survival (OS) and disease-free survival (DFS) (*p* = 0.02, 0.01, respectively). Conversely, patients with a decrease in both NLR and PLR, showed the most favorable survival outcomes. These findings suggest that changes in NLR may have a stronger prognostic influence on survival than changes in PLR alone; however, the dynamic changes in both markers may still be considered when stratifying patients after CRT.

In recent years, increasing evidence has linked colorectal cancer (CRC) with systemic and local inflammatory responses [28]. Inflammatory biomarkers such as the neutrophil-to-lymphocyte ratio (NLR) and platelet-to-lymphocyte ratio (PLR) have been studied for their association with disease progression and survival [29,30]. Several studies have supported the prognostic role of these markers in CRC, particularly in relation to tumor stage, treatment response, and recurrence risk [26,31,32,33,34]. Elevated pre-treatment NLR levels have been associated with more advanced disease and poorer survival outcomes, including overall and disease-specific survival, particularly in stages II and III CRC [35].

Our findings are consistent with previous studies demonstrating that NLR is a more robust prognostic biomarker than PLR in patients with rectal cancer. A recent retrospective study by El Mohtaseb et al. in colorectal cancer patients reported that high preoperative NLR was significantly associated with poor overall survival (OS) and cancer-specific survival (CSS). Notably, this association persisted even after adjustment for clinical confounders. In contrast, no significant association was observed between PLR and OS, further supporting the limited prognostic utility of PLR relative to NLR [36]. These findings concur with our results, in which only post-CRT NLR retained significance in multivariable survival models, while PLR failed to demonstrate an independent association with either OS or DFS.

Our findings are supported by a study conducted by Cha et al., which reported that persistent low NLR during the course of chemoradiotherapy was associated with improved disease-free survival (DFS). Whereas individual pre- and post-CRT NLR measurements alone were not predictive of survival [37]. This suggests prolonged inflammatory suppression that occurs during treatment, and its associated dynamic changes have a greater prognostic value than the standard, baseline measurements. Jeon et al. reported that an elevated post-CRT NLR (>5.21) was associated with decreased recurrence-free survival, and further aided the prognostic importance of post-treatment inflammatory markers [38]. Similarly, Sung et al. demonstrated that a post-CRT NLR greater than 5.14 was independently associated with poorer DFS and OS [39]. Most recently, Morais et al. demonstrated that a high post-CRT NLR (>4) was associated, independently, with worse overall survival after a multivariate analysis, with Kaplan–Meier curves further validating this value [40]. These studies collectively support our findings, highlighting post-treatment NLR as a meaningful biomarker for outcomes.

While our findings suggest a limited prognostic role for PLR in rectal cancer, the literature on this biomarker remains inconsistent. Several studies have reported a significant association between elevated PLR and poor survival outcomes, particularly regarding disease-specific survival or loco-regional control. For example, Partl et al. found that pre-treatment PLR was a significant prognostic factor for loco-regional control in patients with locally advanced rectal cancer undergoing neoadjuvant therapy [41]. Similarly, a systematic review and meta-analysis by Tan et al. confirmed that elevated PLR was associated with worse overall and disease-free survival in colorectal cancer patients [42]. Furthermore, Jia et al. studied 145 patients with colorectal cancer who had undergone neoadjuvant chemotherapy between January 2011 and February 2014, of whom 100 were rectal cancer cases. They found that PLR, was associated with OS and DFS in their univariable analysis of CRC patients [33]. These findings suggest that the prognostic utility of PLR may vary depending on treatment modality, disease stage, and biomarker timing.

Our findings differ from those of the meta-analysis of Lee et al., which identified elevated pre-CRT NLR as a significant predictor of worse overall survival (OS) and disease-free survival (DFS) in rectal cancer patients undergoing preoperative CRT [43]. They highlighted the role of pre-treatment systemic inflammation in shaping long-term outcomes. However, in contrast, our study found no significant association between pre-CRT inflammatory markers and survival outcomes. This discrepancy likely reflects the dynamic changes in NLR levels observed in our subgroup analysis. Specifically, we found that a high pre-CRT NLR followed by a decrease in post-CRT was associated with better survival outcomes, whereas patients with persistently high NLR levels had worse results.

This reinforces the idea that the direction of NLR change is more prognostically informative than baseline values alone. Our findings are supported by Liu et al., who investigated NLR dynamics in metastatic colorectal cancer patients treated with FOLFOX and targeted agents. Their study found that patients with high baseline NLR who experienced a subsequent drop after treatment had significantly improved progression-free survival (PFS), compared to those with persistently elevated NLR levels [44]. This suggests that the change in NLR, rather than its baseline value alone, plays a more crucial role in predicting patient outcomes.

Together with our data, these studies reinforce the dominant prognostic role of NLR over PLR in rectal cancer and support its clinical integration into post-CRT risk stratification. Importantly, our study also adds the novel dimension of post-treatment biomarker dynamics, demonstrating that changes in NLR (rather than absolute values) further enhance prognostic accuracy and better reflect treatment response.

In addition to the NLR levels, CEA levels were independently associated with overall survival (OS), even after adjusting for established clinical variables such as age, comorbidity, and AJCC pathologic stage. This suggests that these markers may capture elements of residual tumor burden and/or systemic inflammatory response that are not fully accounted for by conventional staging systems. Notably, patients with elevated post-CRT NLR had a 3.6-fold increased risk of mortality, while elevated post-CRT CEA was associated with more than a 15-fold increased risk—underscoring the powerful prognostic value of these markers. These results align with prior literature linking elevated NLR and CEA to poor outcomes in rectal cancer [40,45,46,47].

Our findings are consistent with a meta-analysis by Tsai et al., which included 15 studies and evaluated over 7700 colorectal cancer patients, of whom 1391 (31%) had non-metastatic rectal cancer. The analysis reported that both a low NLR (<5) and CEA level (<5 ng/mL) were significantly associated with better 5-year overall and disease-free survival. Furthermore, elevated NLR was correlated with adverse tumor features such as larger size, poorer differentiation, and higher CEA levels [26]. These results underscore the biological and clinical rationale for combining these two routinely available markers to enhance prognostication.

However, our findings extend this evidence by demonstrating their additive predictive value when assessed together after CRT, where the strength of combining NLR and CEA was further supported by our ROC analysis, which showed that while each marker had moderate predictive value alone (AUC = 0.66 for NLR, 0.70 for CEA), their combination markedly improved prognostic accuracy (AUC = 0.84). This suggests a complementary biological relationship, in which NLR reflects host immune-inflammation, while CEA represents tumor-specific activity [48]. The combined model may thus serve as a practical and non-invasive tool for identifying high-risk patients who may benefit from intensified surveillance or adjuvant therapy [49,50,51].

Our study also confirmed the established role of pathologic AJCC stage as a powerful independent predictor of both OS and DFS, with stage III disease significantly associated with increased recurrence and mortality, which is well-documented and consistently validated across numerous studies in the rectal cancer literature [52,53]. However, even within the same pathologic stage, inflammatory and tumor markers provided additional prognostic discrimination, supporting their complementary utility.

While our study focused on the prognostic value of NLR, PLR, and CEA, previous research has identified other serum biomarkers, including IL-6, TNF-αR2, adiponectin, and C-reactive protein as potential indicators of prognosis in rectal cancer [54,55,56]. These inflammatory and metabolic markers have been associated with tumor progression, systemic inflammation, and survival outcomes [57,58,59]. Although these biomarkers were not routinely measured in our cohort and were therefore beyond the scope of this retrospective study, their inclusion in future studies could provide additional insights into patient stratification and complement the predictive value of NLR and CEA. Prospective studies incorporating a broader panel of biomarkers may help refine prognostic models and guide personalized treatment strategies.

### Limitations

This study has several limitations. Firstly, the study being a retrospective study in a single-institution setting may limit generalizability. Additionally, while we used robust multivariable models, residual confounding cannot be excluded. MMR, RAS, and BRAF molecular status were not routinely available for the included patients and therefore could not be analyzed; future studies incorporating these molecular biomarkers may provide additional prognostic insights. Detailed information on concomitant medications before, during, and after neoadjuvant chemotherapy was also not available; however, prior evidence suggests that such factors are unlikely to have substantially affected the prognostic value of NLR and PLR. Finally, while our findings support the clinical relevance of NLR and CEA, future prospective studies are needed to validate their predictive value and determine how they may be used to guide treatment decisions, such as selecting patients for intensified adjuvant therapy or immunotherapy or surveillance.

Despite these limitations, a major strength of this study lies in its comprehensive evaluation of both static and dynamic changes in inflammatory (NLR, PLR) and tumor (CEA) markers before and after neoadjuvant chemoradiotherapy in a well-defined cohort of patients with locally advanced rectal cancer and standardized management protocol. In contrast to many previous studies that focus on pre-treatment values alone, we incorporated post-treatment levels and their directional changes (delta values), offering a more nuanced understanding of treatment response and prognosis. Additionally, we assessed the combined prognostic power of NLR and CEA, demonstrating a significant improvement in predictive accuracy when used together, an approach that is clinically relevant and easily applicable using routine blood tests. The study also benefits from multivariable analysis adjusting for key clinical and pathological covariates, enhancing the reliability and generalizability of our findings.

## 5. Conclusions

In conclusion, post-CRT levels and dynamic changes in NLR and CEA serve as independent prognostic markers in LARC cancer and can significantly enhance survival prediction when used in combination. Their incorporation into post-treatment assessment may support risk-adapted follow-up strategies and help identify patients who may benefit from further intervention. Future prospective validation and integration into clinical decision algorithms are warranted.

## Figures and Tables

**Figure 1 cancers-17-03383-f001:**
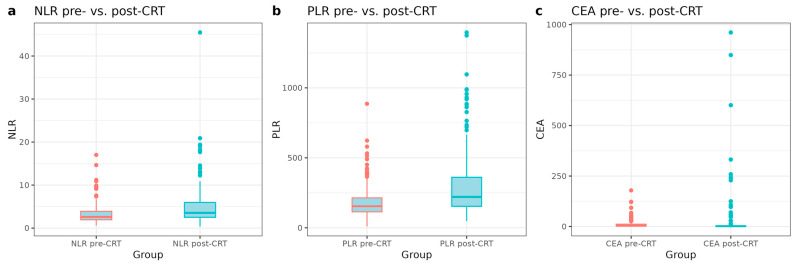
Distribution of NLR, PLR, and CEA before and after chemoradiotherapy (CRT). Box plots showing pre- and post-CRT values of the (**a**) neutrophil-to-lymphocyte ratio (NLR), (**b**) platelet-to-lymphocyte ratio (PLR), and (**c**) carcinoembryonic antigen (CEA). Differences between pre- and post-CRT levels were assessed using paired *t*-tests; *p*-values are indicated.

**Figure 2 cancers-17-03383-f002:**
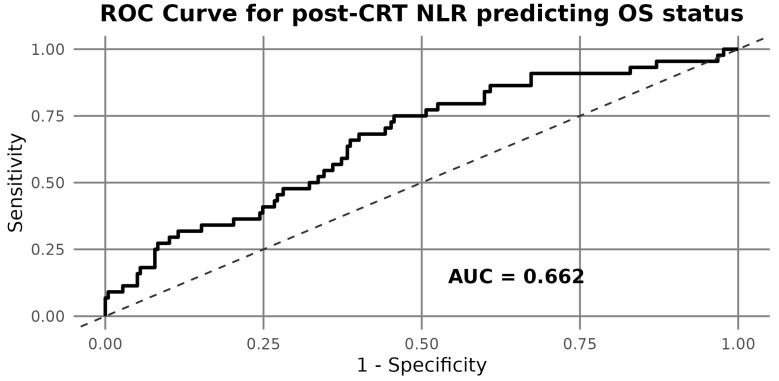
Receiver operating characteristic (ROC) curve of post-CRT NLR for overall survival. The optimal cut-off point was 8.807.

**Figure 3 cancers-17-03383-f003:**
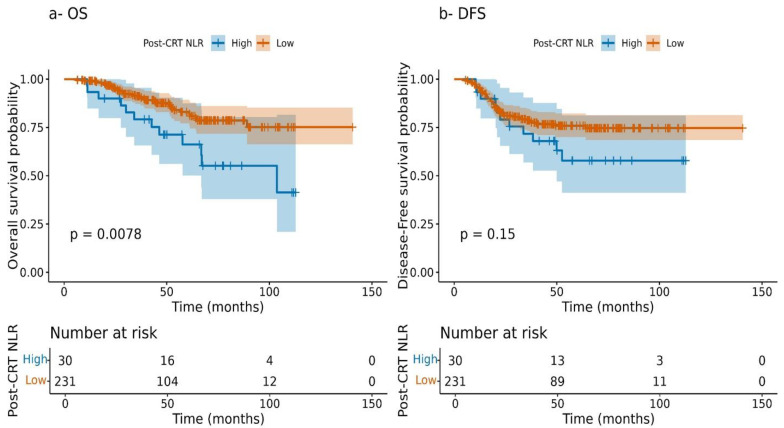
Kaplan–Meier survival curves according to post-CRT NLR. (**a**) Overall survival (OS) and (**b**) disease-free survival (DFS) were stratified by post-CRT NLR and compared using the log-rank test.

**Figure 4 cancers-17-03383-f004:**
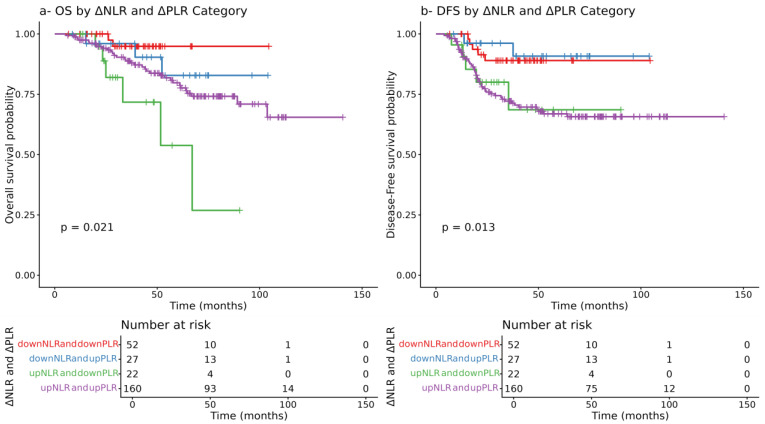
Kaplan–Meier survival curves according to ΔNLR–ΔPLR groups. (**a**) Overall survival (OS) and (**b**) disease-free survival (DFS) were stratified by combined changes in NLR and PLR (ΔNLR–ΔPLR) and compared using the log-rank test.

**Figure 5 cancers-17-03383-f005:**
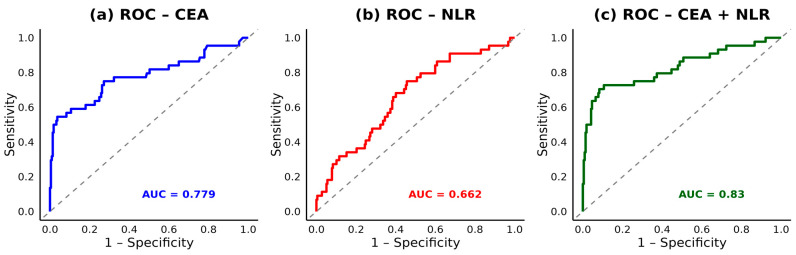
Receiver operating characteristic (ROC) curves of CEA and NLR for overall survival prediction. (**a**) CEA alone, (**b**) NLR alone, and (**c**) CEA combined with NLR.

**Figure 6 cancers-17-03383-f006:**
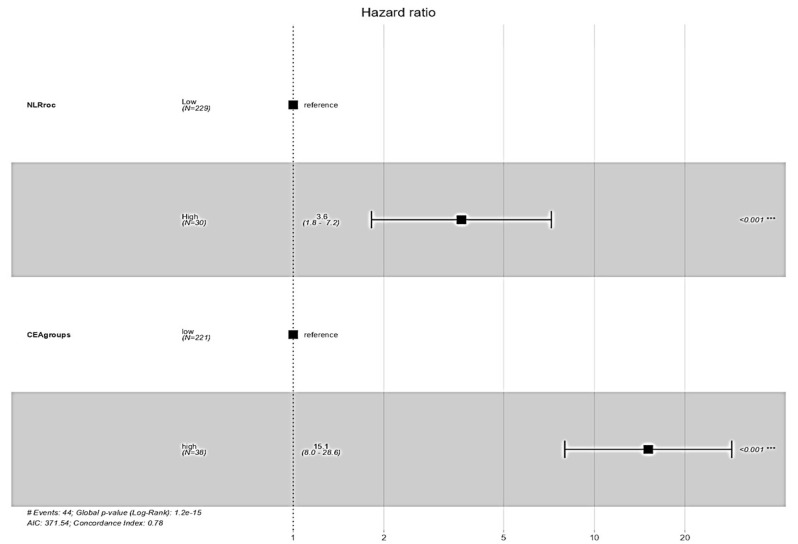
Hazards regression plot showing hazard ratios (HRs) with 95% confidence intervals for overall survival (OS) across NLR and CEA groups. *** indicates a significance level of *p* < 0.001, and # denotes the corresponding number of events.

**Figure 7 cancers-17-03383-f007:**
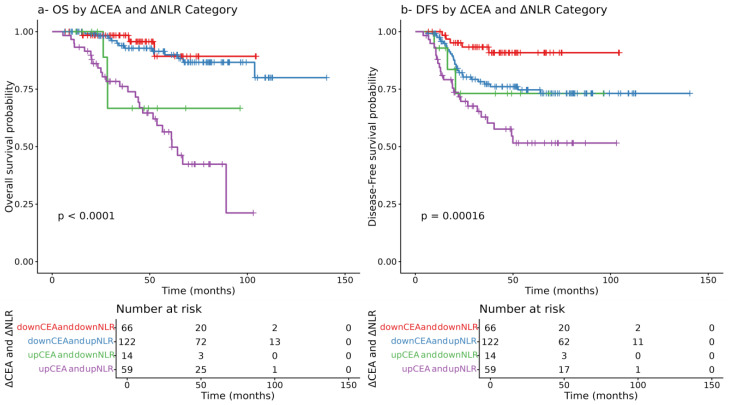
Kaplan–Meier curves of overall survival (OS) and disease-free survival (DFS) according to changes in neutrophil-to-lymphocyte ratio (NLR) and carcinoembryonic antigen (CEA).

**Table 1 cancers-17-03383-t001:** Demographics and clinical characteristics of the participants.

	Overall (N = 261)
Age	
Mean (SD)	55.5 (11.4)
Range	25.0–80.0
Gender	
Female	115 (44.1%)
Male	146 (55.9%)
BMI	
Mean (SD)	29.5 (5.9)
Range	11.6–52.9
Comorbidity	
No	98 (37.5%)
Yes	163 (62.5%)
Smoking	
Yes	98 (37.5%)
no	163 (62.5%)
Clinical Stage—Radiological	
2	17 (6.5%)
3	244 (93.5%)
Therapy group	
CCRT	169 (64.8%)
TNT	92 (35.2%)
Surgery	
APR	71 (27.2%)
LAR	187 (71.6%)
Pelvic Exenteration	3 (1.1%)
Surgical approach	
Minimally invasive	101 (38.3%)
Open	161 (61.7%)
Postoperative pathological AJCC	
0	41 (15.7%)
1	50 (19.2%)
2	95 (36.4%)
3	75 (28.7%)
Distance from anal verge	
N-Miss	4
Mean (SD)	6.6 (3.1)
Range	1.0–17.0
Complication	
No	159 (60.9%)
Yes	102 (39.1%)

**Table 2 cancers-17-03383-t002:** Dynamic changes in inflammatory and tumor markers pre- and post-CRT.

	Pre-CRT	Post-CRT	Paired *t*-Test *p*	Wilcoxon *p*
CEA			0.126	<0.01
Mean (SD)	11.4 (20.2)	20.6 (95.8)		
Median (Range)	4.2 (0.4–179.0)	1.9 (0.2–961.0)		
NLR			<0.001	<0.001
Mean (SD)	3.2 (2.1)	4.9 (4.3)		
Median (Range)	2.6 (0.6–17.0)	3.53 (0.4–45.5)		
PLR			<0.001	<0.001
Mean (SD)	185.4 (110.2)	293.5 (217.6)		
Median (Range)	154 (9.5–886.3)	221 (48.4–1395.9)		

**Table 3 cancers-17-03383-t003:** Univariable and multivariable survival analysis for baseline characteristics.

		Overall Survival		Disease-Free Survival	
		HR (Univariable)	HR (Multivariable)	HR (Univariable)	HR (Multivariable)
Gender	Female	-		-	
	Male	1.24 (0.67–2.27, *p* = 0.495)		1.34 (0.79–2.27, *p* = 0.271)	
Comorbidity	No	-		-	
	Yes	3.49 (1.48–8.27, *p* = 0.004)	2.17 (0.86–5.46, *p* = 0.099)	1.70 (0.96–3.02, *p* = 0.068)	
Smoking	Yes	-		-	
	no	0.99 (0.54–1.84, *p* = 0.978)		1.13 (0.66–1.92, *p* = 0.657)	
Clinical Stage	2	-		-	
	3	0.55 (0.22–1.39, *p* = 0.207)		0.54 (0.23–1.25, *p* = 0.150)	
Treatment Group	CCRT	-			
	TNT	0.7 (0.26–1.88, *p* = 0.484)		0.9 (0.46–1.94, *p* = 0.612)	
Surgery	APR	-		-	
	LAR	0.80 (0.42–1.52, *p* = 0.490)		0.90 (0.51–1.56, *p* = 0.699)	
	Pelvic Exenteration	4.23 (0.96–18.66, *p* = 0.057)		3.29 (0.76–14.20, *p* = 0.110)	
Surgical approach	Minimally invasive	-		-	
	Open	1.32 (0.91–2.64, *p* = 0.721)		1.21 (0.69–2.10, *p* = 0.509)	
Complication	No	-		-	
	Yes	1.68 (0.91–3.10, *p* = 0.097)		1.41 (0.85–2.35, *p* = 0.182)	
AJCC—postoperative pathologic stage	0	-	-	-	-
	1	1.40 (0.34–5.87, *p* = 0.643)	1.38 (0.33–5.79, *p* = 0.658)	1.42 (0.42–4.87, *p* = 0.572)	1.42 (0.41–4.84, *p* = 0.578)
	2	2.33 (0.67–8.05, *p* = 0.181)	1.89 (0.54–6.60, *p* = 0.317)	2.09 (0.71–6.19, *p* = 0.181)	2.05 (0.69–6.06, *p* = 0.194)
	3	3.67 (1.09–12.33, *p* = 0.036)	2.01 (0.57–7.08, *p* = 0.277)	5.10 (1.80–14.46, *p* = 0.002)	4.12 (1.43–11.82, *p* = 0.009)
Age	Mean (SD)	1.03 (1.01–1.06, *p* = 0.011)	1.03 (1.00–1.06, *p* = 0.070)	1.02 (1.00–1.04, *p* = 0.125)	
BMI	Mean (SD)	1.00 (0.95–1.05, *p* = 0.981)		1.00 (0.96–1.05, *p* = 0.845)	
CEA pre-CRT	Mean (SD)	1.00 (0.99–1.01, *p* = 0.640)		1.00 (1.00–1.01, *p* = 0.309)	
CEA post-CRT	Mean (SD)	1.00 (1.00–1.01, *p* < 0.001)	1.00 (1.00–1.01, *p* < 0.001)	1.01 (1.00–1.01, *p* < 0.001)	1.00 (1.00–1.01, *p* < 0.001)
NLR pre-CRT	Mean (SD)	1.04 (0.90–1.21, *p* = 0.592)		0.89 (0.76–1.05, *p* = 0.169)	
PLR pre-CRT	Mean (SD)	1.00 (1.00–1.00, *p* = 0.659)		1.00 (1.00–1.00, *p* = 0.365)	
NLR post-CRT	Mean (SD)	1.06 (1.02–1.10, *p* = 0.003)	1.05 (1.01–1.09, *p* = 0.007)	1.03 (0.98–1.07, *p* = 0.246)	
PLN post-CRT	Mean (SD)	1.00 (0.98–1.00, *p* = 0.674)		1.00 (1.00–1.00, *p* = 0.072)	

## Data Availability

The data that support the findings of this study are available from the corresponding author upon reasonable request.

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
