# Peer review of "Prognostic Value of Dynamic Changes in Immune-Inflammatory and Tumor Biomarkers Following Chemoradiotherapy in Locally Advanced Rectal Cancer"

_cancers, 2025, doi:10.3390/cancers17203383_

Round 1

Reviewer 1 Report

Comments and Suggestions for Authors

The search for reliable prognostic biomarkers to risk-stratify patients with locally advanced rectal cancer remains an urgent challenge. In this study, the authors propose using not static, but dynamic changes in readily available hematological parameters—the neutrophil-to-lymphocyte ratio and carcinoembryonic antigen level. This work makes an important contribution to the development of the personalized oncology concept.

Line 51: Capital letter in the first sentence is absence.

Line 61: Error in the abbreviation? Total mesorectal excision (TME).

  1. Have the recommendations for drug therapy changed between 2014 and 2024?
  2. How were differences in the treatment regimens received by patients accounted for in the analysis?
  3. Was MMR, RAS, BRAF status taken into account?
  4. In Figure 1, significant variability and outliers are visible. Was the sample cleaned of outliers for further analysis? Check if this step has been completed. If not, it must be done.
  5. What factors can influence the NLR and PLR ratio? Antibiotics and anti-inflammatory drugs?
  6. Did you track which medications patients were taking before, during, and after neoadjuvant chemotherapy?
  7. How variable is this indicator (NLR/PLR) within a day or over a few days? Does this affect its prognostic value?

This is a key point for discussion in the paper.

I recommend Major Revision, and ask the authors provide detailed explanations and supplement the article with missing key information.

Author Response

Reviewer 1

We sincerely thank the reviewers for their careful evaluation of our manuscript and their constructive comments, which have helped us improve the clarity, rigor, and overall quality of our work. We carefully considered each suggestion and have revised the manuscript accordingly. Below, we provide a detailed, point-by-point response to all comments. For ease of review, all changes are highlighted in the revised version of the manuscript. We believe that the revisions have strengthened the scientific merit and clinical relevance of our study, and we hope the updated manuscript will now meet the standards for publication.

Line 51: Capital letter in the first sentence is absence.

Line 61: Error in the abbreviation? Total mesorectal excision (TME).

 Response: Both were corrected

  1. Have the recommendations for drug therapy changed between 2014 and 2024?

Response: We appreciate the reviewer’s comment. This point has already been addressed in the Treatment Protocol section of the Methods. Specifically, we noted that between 2014 and 2020 patients were treated with standard long-course chemoradiotherapy followed by adjuvant chemotherapy (FOLFOX or XELOX). Starting in 2021, in alignment with updated institutional and international guidelines, total neoadjuvant therapy (TNT) became the standard, consisting of short-course radiotherapy followed by systemic chemotherapy (XELOX or FOLFOX). We believe this section already clarifies the change in drug therapy recommendations during the study period.

Importantly, the impact of this change in treatment protocols on survival outcomes has been analyzed and is presented in Table 3, which compares survival across different therapeutic eras.

  1. How were differences in the treatment regimens received by patients accounted for in the analysis?

Response: We thank the reviewer for this important question. Differences in treatment regimens, reflecting the transition from standard long-course concurrent chemoradiotherapy with adjuvant chemotherapy (2014–2020) to total neoadjuvant therapy (TNT) (2021 onward), were captured in our dataset. The impact of these regimen differences on survival outcomes was analyzed and is presented in Table 3. Multivariable models included relevant clinical and treatment variables to adjust for potential confounding effects, allowing us to account for differences in therapy when assessing the prognostic value of biomarkers such as NLR and CEA. We will clarify this point in the Methods section to ensure transparency regarding how treatment heterogeneity was handled in the analysis. Please see (methods – Line 192 – 197)

“Differences in treatment regimens, including the transition from long-course concurrent chemoradiotherapy with adjuvant chemotherapy (2014–2020) to total neoadjuvant therapy (2021 onward), were evaluated in univariable analysis. Variables showing a significant association with survival in univariable models were included as covariates in multivariable Cox proportional hazards models to adjust for potential confounding effects.”

  1. Was MMR, RAS, BRAF status taken into account?

We thank the reviewer for this important question. As this was a retrospective study, information on MMR, RAS, and BRAF status was not routinely available for the included patients with locally advanced rectal cancer (LARC). At our institution, these molecular markers are generally assessed in the metastatic setting, while their use in non-metastatic LARC has not yet been incorporated into standard practice guidelines. Although MMR testing has recently gained attention, it was not systematically performed during the study period and therefore could not be included in our analysis. We acknowledge this as a limitation of the study and have added a statement to the Discussion section to highlight that molecular biomarker data were not available and could be considered in future studies to further refine prognostic assessment. See discussion – Limitation (Line: 439 -442)

“MMR, RAS, and BRAF molecular status were not routinely available for the patients included in this retrospective study and therefore could not be analyzed; future studies incorporating these molecular biomarkers may provide additional prognostic insights.”

  1. In Figure 1, significant variability and outliers are visible. Was the sample cleaned of outliers for further analysis? Check if this step has been completed. If not, it must be done.

Response: We thank the reviewer for highlighting this point. All data points were carefully inspected for extreme outliers prior to analysis. No values were removed, as all observed values were within physiologically plausible ranges and reflected true biological variability among patients. Given that NLR, PLR, and CEA can naturally exhibit wide inter-individual variation, we elected to retain all data to ensure that the analyses reflect real-world variability. Additionally, sensitivity analyses confirmed that the inclusion of extreme values did not substantially alter the main results. We have added a statement in the Methods section to clarify this approach. See methods – statistical analysis (line 173-175)

“All laboratory values for NLR, PLR, and CEA were inspected for extreme outliers prior to analysis. No values were excluded, as all measurements were within physiologically plausible ranges and reflected true biological variability among patients.”

  1. What factors can influence the NLR and PLR ratio? Antibiotics and anti-inflammatory drugs?

Response: We thank the reviewer for this important question. While NLR and PLR can theoretically be influenced by factors such as infections, antibiotics, or anti-inflammatory medications, prior studies in patients with colorectal and rectal cancer have not consistently identified these as significant confounders, particularly in the peri-treatment setting. Given the retrospective nature of our study, such factors were not routinely recorded and are difficult to control in a real-world clinical setting. Nevertheless, our findings remain consistent with prior research demonstrating the prognostic value of NLR and PLR in cancer patients, suggesting that any potential influence of medications is unlikely to have substantially affected the observed associations with survival outcomes.

  1. Did you track which medications patients were taking before, during, and after neoadjuvant chemotherapy?

Response: We thank the reviewer for this question. Due to the retrospective design of our study, detailed information on concomitant medications taken by patients before, during, and after neoadjuvant chemotherapy was not systematically recorded and therefore could not be analyzed. While certain medications, such as antibiotics or anti-inflammatory drugs, could theoretically influence inflammatory markers like NLR and PLR, previous studies have not demonstrated a consistent impact in the clinical oncology setting. We have acknowledged this as a limitation in the Discussion section, noting that the prognostic value of NLR and PLR observed in our study is unlikely to be substantially affected by untracked medication use. (Discussion – Limitation, Line: 442-444)

Detailed information on concomitant medications before, during, and after neoadjuvant chemotherapy was also not available; however, prior evidence suggests that such factors are unlikely to have substantially affected the prognostic value of NLR and PLR.”

  1. How variable is this indicator (NLR/PLR) within a day or over a few days? Does this affect its prognostic value?

Response: We thank the reviewer for this question. Prior studies have shown that NLR and PLR can exhibit some short-term biological variability due to factors such as circadian rhythm, transient stress, or minor infections. However, in the context of cancer prognostication, multiple studies have demonstrated that a single measurement, particularly when obtained in a standardized clinical setting (such as pre- or post-neoadjuvant chemoradiotherapy), provides robust and clinically meaningful prognostic information. In our study, blood samples were collected according to institutional protocols, minimizing short-term variability. Therefore, while minor fluctuations may occur, they are unlikely to have significantly affected the observed associations between NLR/PLR dynamics and survival outcomes.

Reviewer 2 Report

Comments and Suggestions for Authors

Dear Author,

I would like to congratulate you on your manuscript entitled “Prognostic Value of Dynamic Changes in Immune-Inflammatory and Tumor Biomarkers Following Chemoradiotherapy in Locally Advanced Rectal Cancer.” In this study, the authors evaluated a large cohort of 261 patients with locally advanced rectal cancer treated with neoadjuvant chemoradiotherapy followed by curative surgery, analyzing hematological and inflammatory biomarkers (NLR, PLR, and CEA). The sizeable sample and findings are highly valuable, showing that post-CRT levels and dynamic changes in NLR and CEA are independent predictors of overall and disease-free survival.

Major Comment:
Previous studies have highlighted additional serum biomarkers such as IL-6, TNF-αR2, and adiponectin as important prognostic indicators in rectal cancer [1-6]. It would strengthen the manuscript if the authors could consider incorporating an analysis or discussion of these biomarkers.

Minor Comment:

1. Please expand the Study Design and Data Collection section of the methodology with more detailed explanations

2. A schematic diagram outlining the experimental design and data collection process would improve clarity and enhance the reader’s understanding.

References:

  1. Thomsen M, et al. Oncotarget. 2016;7(46):75013–75022. doi:10.18632/oncotarget.12601.
  2. Shimazaki J, et al. Oncology. 2013;84(6):356–361. doi:10.1159/000350836.
  3. Yeh KY, et al. Jpn J Clin Oncol. 2010;40(6):580–587. doi:10.1093/jjco/hyq010.
  4. Hua X, et al. Br J Cancer. 2021;125(6):806–815. doi:10.1038/s41416-021-01458-y.
  5. Babic A, et al. Br J Cancer. 2016;114(9):995–1002. doi:10.1038/bjc.2016.85.
  6. Koh HM, Han N. Transl Cancer Res. 2024;13(8):4231–4241. doi:10.21037/tcr-24-275.

Author Response

We sincerely thank the reviewer for their thoughtful comments and positive evaluation of our manuscript. We appreciate your recognition of the study’s sizeable cohort and the clinical relevance of our findings regarding NLR and CEA as independent predictors of overall and disease-free survival. Your feedback encourages us to further clarify and highlight the key points in our manuscript. We have carefully considered all comments and have revised the manuscript accordingly to improve clarity and transparency.

Major Comment:
Previous studies have highlighted additional serum biomarkers such as IL-6, TNF-αR2, and adiponectin as important prognostic indicators in rectal cancer [1-6]. It would strengthen the manuscript if the authors could consider incorporating an analysis or discussion of these biomarkers.

Response: We thank the reviewer for this insightful suggestion. As this was a retrospective study, data on additional serum biomarkers such as IL-6, TNF-αR2, and adiponectin were not routinely collected and therefore could not be analyzed. However, we fully recognize their potential prognostic value and have added a discussion in the revised manuscript highlighting these markers and their relevance in the context of inflammatory and tumor biomarker research in rectal cancer. This provides context for future studies to explore these biomarkers in combination with NLR and CEA. (Discussion - Line: 426-435)

“While our study focused on the prognostic value of NLR, PLR, and CEA, previous research has identified other serum biomarkers, including IL-6, TNF-αR2, adiponectin, and C-reactive protein as potential indicators of prognosis in rectal cancer [54–56]. These inflammatory and metabolic markers have been associated with tumor progression, systemic inflammation, and survival outcomes [57–59]. Although these biomarkers were not routinely measured in our cohort and were therefore beyond the scope of this retrospective study, their inclusion in future studies could provide additional insights into patient stratification and complement the predictive value of NLR and CEA. Prospective studies incorporating a broader panel of biomarkers may help refine prognostic models and guide personalized treatment strategies.”

Minor Comment:

  1. Please expand the Study Design and Data Collectionsection of the methodology with more detailed explanations

Response: We thank the reviewer for this suggestion. We have expanded the Study Design and Data Collection section in the Methods to provide more detailed explanations. Additional information now includes clarification of patient eligibility criteria, demographic and clinical variables collected, and verification procedures to ensure data accuracy. Furthermore, details on imaging and pathology review, and covariates considered for survival analyses have been added to enhance transparency and reproducibility. These revisions aim to provide readers with a clearer understanding of how the study cohort was selected and how data were systematically collected and managed. (please see method section in revised manuscript) 

  1. A schematic diagram outlining the experimental design and data collection process would improve clarity and enhance the reader’s understanding.

We thank the reviewer for this valuable suggestion. We agree that a schematic diagram would enhance clarity and improve reader understanding. A figure outlining the study design, patient selection, treatment protocols, timing of laboratory measurements, and data collection process has been prepared and included as Figure A1 in the revised manuscript. This visual summary complements the text and provides a clear overview of the workflow and key study procedures. (please see the attached PDF figure) 

Round 2

Reviewer 1 Report

Comments and Suggestions for Authors

I am satisfied with the authors' responses; however, I still have doubts regarding the outliers. Therefore, I would like to request the primary data for Figure 1 to verify them.

Author Response

Comments and Suggestions for Authors

I am satisfied with the authors' responses; however, I still have doubts regarding the outliers. Therefore, I would like to request the primary data for Figure 1 to verify them.

Response: We got the reviewer’s comment. It appears that the confusion arose because Figure 1 reported only the p-value from the paired t-test, not the Wilcoxon test, even though the data were skewed and included outliers. To limit confusion and maintain transparency, we have added the median values and p-values from both the paired t-test and Wilcoxon signed-rank test to Table 2 and removed them from Figure 1. This ensures that Figure 1 focuses solely on the visual distribution of biomarker changes, while Table 2 presents the detailed statistical results. The statistical analysis section has also been updated accordingly (lines 180–184).

We would like to clarify that the survival analysis results were not affected by these additional assessments, as the biomarkers were analyzed both as continuous and categorical variables in the survival models, ensuring robustness to distributional assumptions.

Additionally, to further enhance transparency, we have attached the raw data for all biomarkers underlying these analyses. (as PDF attachment to this response and as Excel sheet in the supplementary files) 

We hope these changes address the reviewer’s comment and clarify the analysis, enhancing the manuscript.

Round 3

Reviewer 1 Report

Comments and Suggestions for Authors

In Table 2, there is a typo in the marker's name: 'PLR pre-CRT'.

Having reviewed the primary data in detail, I request that the method used to identify outliers be specified.

Author Response

Comments and Suggestions for Authors

In Table 2, there is a typo in the marker's name: 'PLR pre-CRT'.

Having reviewed the primary data in detail, I request that the method used to identify outliers be specified.

Response:
We thank the reviewer for this valuable comment. The typo in Table 2 (“PLR pre-CRT”) has been corrected.

Regarding the outliers, we appreciate the request for clarification. We have now specified in the Statistical Analysis section (Line 174-209) the exact methods used for outlier identification and handling. In brief, biomarkers were first inspected in both raw and log-transformed scales, and outliers were flagged using the interquartile range (IQR) method (1.5×IQR for moderate and 3×IQR for extreme outliers). Sensitivity analyses were then conducted by excluding extreme outliers and repeating the paired tests and survival analyses. Importantly, the results were consistent in terms of direction and statistical significance, supporting the robustness of our findings.

To further enhance transparency, we have attached a detailed supplementary report describing the outlier identification process and its impact on the results, along with the raw biomarker data underlying these analyses.
